# Sourdough Fermented Breads are More Digestible than Those Started with Baker’s Yeast Alone: An In Vivo Challenge Dissecting Distinct Gastrointestinal Responses

**DOI:** 10.3390/nu11122954

**Published:** 2019-12-04

**Authors:** Carlo Giuseppe Rizzello, Piero Portincasa, Marco Montemurro, Domenica Maria Di Palo, Michele Pio Lorusso, Maria De Angelis, Leonilde Bonfrate, Bernard Genot, Marco Gobbetti

**Affiliations:** 1Department of Soil, Plant and Food Science, University of Bari Aldo Moro, 70126 Bari, Italy; montemarco@yahoo.it (M.M.);; 2Clinica Medica “A. Murri”, Department of Biomedical Sciences & Human Oncology, University of Bari Medical School, 70124 Bari, Italy; 3Puratos NV, 1702 Groot-Bijgaarden, Belgium; 4Faculty of Science and Technology, Free University of Bozen-Bolzano, 39100 Bolzano, Italy; marco.gobbetti@unibz.it

**Keywords:** gallbladder emptying, hydrogen breath test, stomach emptying, orocecal transit time, test meal, ultrasonography

## Abstract

As a staple food, bread digestibility deserves a marked nutritional interest. Combining wide-spectrum characterization of breads, in vitro nutritional indices, and in vivo postprandial markers of gastrointestinal function, we aimed at comparing the digestibility of sourdough and baker’s yeast breads. Microbiological and biochemical data showed the representativeness of the baker´s yeast bread (BYB) and the two sourdough breads (SB and t-SB, mainly differing for the time of fermentation) manufactured at semi-industrial level. All in vitro nutritional indices had the highest scores for sourdough breads. Thirty-six healthy volunteers underwent an in vivo challenge in response to bread ingestion, while monitoring gallbladder, stomach, and oro-cecal motility. SB, made with moderate sourdough acidification, stimulated more appetite and induced lower satiety. t-SB, having the most intense acidic taste, induced the highest fullness perception in the shortest time. Gallbladder response did not differ among breads, while gastric emptying was faster with sourdough breads. Oro-cecal transit was prolonged for BYB and faster for sourdough breads, especially when made with traditional and long-time fermentation (t-SB), whose transit lasted ca. 20 min less than BYB. Differences in carbohydrate digestibility and absorption determined different post-prandial glycaemia responses. Sourdough breads had the lowest values. After ingesting sourdough breads, which had a concentration of total free amino acids markedly higher than that of BYB, the levels in blood plasma were maintained at constantly high levels for extended time.

## 1. Introduction

Bread has been and is one of the worldwide-consumed staple foods [1]. A bread recipe comprises cereal flour (e.g., wheat, rye), even if the use of pseudo-cereals and/or legumes is increasing, tap water, eventually salt and other minor ingredients, and a leavening agent, responsible for gas release and consequently dough expansion. The choice of the leavening agent is crucial for a number of technological factors (e.g., costs and time of processing) and, primarily, because of the repercussions on the bread sensory, texture, nutritional and shelf life features. Commonly, three types of leavening agents may be used: chemicals, baker’s yeast (e.g., commercial preparations of *Saccharomyces cerevisiae* cells), and sourdough. While the production and commercialization of the first two types takes place at industrial levels, the sourdough is an old example of natural starter, the artisanal use of which complies with regional traditions and cultural heritages. Sourdough is a mixture of flour, water and other ingredients (e.g., salt), which is fermented by naturally occurring lactic acid bacteria and yeasts that propagate during back slopping, a traditional procedure in which the sourdough from the previous fermentation cycle is used as the starter to ferment a new mixture of flour and water [2].

Lactic acid bacteria and yeasts come from flour and house microbiota, but their shaping and assembly into a mature biota during fermentation depend on many drivers, such as the chemical and enzyme composition of the flour, and the temperature, redox potential, water content and duration of the process [3]. Lactic acid bacteria dominate the mature sourdough, reaching a cell density >10^8^ cfu/g [4]. The number of yeasts is one/two logarithmic cycles lower [5]. In the second half of the last century, fast leavening processes by chemicals and/or baker’s yeast almost replaced the use of sourdough. Using these leavening agents, the main polymeric cereal components (e.g., proteins, starch) undergo very mild or absent hydrolysis during processing. Such biochemical events deeply affect the bioavailability of nutrients and lead to the release of bioactive and aroma compounds [6,7,8]. A renewed scientific interest in sourdough has developed during the last decades. The Scopus database (November 2019) reports approximately 1400 published items on sourdough biotechnology during the last fifteen years. Compared to the other leavening agents, the sourdough has the capability to positively influence the bread sensory, texture, nutritional and shelf-life features [7,8]. The effects of sourdough fermentation are related to organic acids synthesis, the activation of the endogenous enzymes of the flour as well as the synthesis of microbial secondary metabolites. Among the main advantages related to sourdough use, the increase of the in vitro protein digestibility and amount of soluble fibre, and the decrease of the glycaemic index, phytate content, trypsin inhibitors, and other anti-nutritional factors, have been described [9]. These scientific evidences promoted its technology transfer and favoured an increasing global manufacture of sourdough breads, which reflect the ancient tradition and satisfy the consumer expectations for natural, high nutritious and sustainable foods [10,11]. The current European manufacture of sourdough breads covers 30–50% of the global market and strengthens this renewed interest by industries, artisans and consumers [12]. 

Traditional, empirical and in vitro scientific results all agree that sourdough and, more in general, the long-time fermentation processes are associated with an improved bread digestibility. Nevertheless, to define univocally the concept of bread digestibility from a nutritional perspective seems controversial. Indeed, bread digestibility relies on factors of different nature: the perception of appetite, satiety and gastrointestinal symptoms after ingestion [13,14,15], and the bioavailability of proteins and starch. A few in vivo clinical studies have already demonstrated the effect of sourdough fermentation on starch digestibility. Compared to breads leavened with chemicals or baker’s yeast, the digestible starch fraction of sourdough breads significantly decreased [16,17]. On the contrary, only in vitro data have accumulated for protein digestibility [18]. An in vivo challenge focusing the overall sourdough bread digestibility, which strengthens or not the empirical and the in vitro scientific evidences, is still missing. 

First, breads obtained by baker’s yeast or sourdough fermentation were subjected to an extensive in vitro biochemical characterization. Then, an in vivo clinical study based on complementary techniques of investigation to assess the effect of fermentation on bread digestibility was carried out. The study comprised gastrointestinal motility in response to different breads, and the comparison with a reference test meal. 

## 2. Materials and Methods 

### 2.1. Microorganisms and Flour

*Lactobacillus plantarum* CR1, *Lactobacillus rossiae* CR5 and *Saccharomyces cerevisiae* E10, belonging to the Culture Collection of the Department of Soil, Plant and Food Sciences (University of Bari, Italy), were inoculated into the wheat flour dough used for preparing the sourdough. Lactic acid bacteria were propagated for 24 h at 30 °C on modified MRS broth (Oxoid, Basingstoke, Hampshire, United Kingdom), with the addition of fresh yeast extract (5%, *v*/*v*) and 28 mM maltose, at the final pH of 5.6 (mMRS). The propagation of *S. cerevisiae* E10 was at 30 °C for 48 h on Sabouraud dextrose broth (Oxoid). When used for fermentation, cells of lactic acid bacteria and yeasts were cultivated until reaching the late exponential phase of growth (ca. 12 and 24 h, respectively). Cell suspensions for the inoculum were prepared according to Rizzello et al. [19]. 

The gross chemical characteristics of the wheat (*Triticum aestivum* cv. Appulo) flour used for making sourdoughs and breads were as follows: moisture, 13.2%; protein, 9.5%; fat, 1.2%; ash, 0.4%; and total carbohydrates, 73.5%. 

### 2.2. Sourdough Preparation

A type I sourdough was made and propagated through a traditional protocol [20,21], starting from a fermented dough inoculated with *Lb. plantarum* CR1, *Lb. rossiae* CR5 and *S. cerevisiae* E10. Wheat flour was mixed with tap water, containing cell suspensions, at 60 rpm for 5 min with a IM 5-8 high-speed mixer (Mecnosud, Flumeri, Italy), and the dough (Dough Yield (DY) = (dough/flour weight) × 100 = of 160) was incubated at 30 °C for 16 h. The inoculum corresponded to ca. 5 × 10^7^ and ca. 5 × 10^6^ cfu/g for lactic acid bacteria and yeasts, respectively. Further the first fermentation, four back slopping steps (refreshments) were carried out, mixing 20% of the previously fermented dough with flour and water (DY of 160), and incubating for 8 h at 30 °C. After each fermentation, doughs were stored at 4 °C until the next refreshment. The value of pH of the doughs was determined by a pHmeter (Model 507, Crison, Milan, Italy) with a food penetration probe. Total titratable acidity (TTA) was determined after homogenization of 10 g of dough with 90 mL of distilled water and expressed as the amount (mL) of 0.1 M NaOH required to neutralize the solution, using phenolphthalein as indicator (official American Association for Clinical Chemistry: AACC method 02-31.01). The rate of volume increase of doughs was determined as described by Minervini et al. [22]. After four refreshments, the acidification rate and volume increase of the dough were stable, and the mature sourdough (S) was used for preparing the sourdoughs to be used in bread making. In detail, S_4_ and S_24_ were prepared after refreshing (mixing 20% of S with flour and water, DY of 160) at 30 °C for 4 and 24 h, respectively.

### 2.3. Sourdoughs Characterization

For microbiological analyses, 10 g of each sourdough were homogenized with 90 mL of sterile peptone water (1% (*w*/*v*) of peptone and 0.9% (*w*/*v*) of NaCl) solution. Presumptive lactic acid bacteria were enumerated on MRS (Oxoid) supplemented with cycloheximide (0.1 g liter). Plates were incubated at 30 °C for 48 h, under anaerobiosis (AnaeroGen and AnaeroJar, Oxoid). Yeasts were enumerated on Sabouraud dextrose agar (SDA) (Oxoid) medium supplemented with chloramphenicol (0.1 g/L) at 30 °C for 48 h.

Water/salt-soluble extracts (WSE) of sourdoughs were prepared according to Weiss et al. [23] and used to analyse free amino acids (FAA) and organic acids. FAA were analysed by a Biochrom 30 series Amino Acid Analyzer (Biochrom Ltd., Cambridge Science Park, England) with a Na-cation-exchange column (20 by 0.46 cm internal diameter), as described by Rizzello et al. [19]. Organic acids were determined by High Performance Liquid Chromatography (HPLC), using an ÄKTA Purifier system (GE Healthcare, Buckinghmshire, UK) equipped with an Aminex HPX-87H column (ion exclusion, Biorad, Richmond, CA, USA), and an UV detector operating at 210 nm. Elution was at 60 °C, with a flow rate of 0.6 mL/min, using H_2_SO_4_ 10 mM as mobile phase [24]. The Fermentation Quotient (FQ) was determined as the molar ratio between lactic and acetic acids. Fermentations were carried out in triplicate and each one was analysed in duplicate.

### 2.4. Bread Making

Industrial breads were manufactured at the pilot plant of ValleFiorita s.r.l. (Ostuni, Italy). Three types of bread (ca. 500 g each one, Appendix A) were manufactured. BYB was made mixing (60 × *g* for 5 min, IM 5-8 high-speed mixer, Mecnosud, Flumeri, Italy) wheat flour (62.5% *w*/*w*), water (37.5% *w*/*w*) and 1.5% (*w*/*w*) of baker’s yeast. The fermentation lasted 2 h at 30 °C. SB was manufactured according to a two-stage protocol, which is routinely used in artisanal and industrial bakeries [25]. The formula consists of 20% (*w*/*w*) sourdough S_4_ (fermented for 4 h at 30 °C, step I), which was then mixed (see above) with flour (50% *w*/*w*), water (30% *w*/*w*) and 1.5% baker’s yeast, and further incubated for 1.5 h at 30 °C (step II). The third type of bread was the most traditional (t-SB). It was manufactured according to the above two-stage protocol but without baker’s yeast. The formula consists of 20% (*w*/*w*) sourdough S_24_ (fermented for 24 h at 30°C, step I), which was then mixed (see above) with flour (50% *w*/*w*) and water (30% *w*/*w*), and further incubated for 4 h at 30 °C (step II). The most common percentage (20%, *w*/*w*) used at industrial/artisanal levels was chosen to inoculate S_4_ and S_24_ [25,26]. All breads were baked at 220 °C for 30 min (Combo 3, Zucchelli, Verona, Italy). Bread making was carried out in triplicate and each bread was analysed twice.

### 2.5. Biochemical, Textural and Nnutritional Characterization of Breads

The values of pH, TTA, organic acids and FAA were determined on fermented doughs before baking, as described elsewhere. Gluten content was determined on doughs before baking by using Glutomatic 2200 following AACC Method 38-12. Protein (total nitrogen × 5.7), lipids and ash were determined according to the AACC official methods 46-11A, 30-10.01, and 08-01, respectively while total carbohydrates were calculated as the difference (100 − (proteins + lipids + ash)). The determination of dietary fibre was carried out by Association of Official Analytical Chemists (AOAC) approved methods 991.43. Energy value was determined according to FAO guidelines [27]. Breads proximal composition and energy value are reported in Appendix A.

Instrumental Texture Profile Analysis (TPA) was carried out with a TVT-300XP Texture Analyser (TexVol Instruments, Viken, Sweden), equipped with a cylinder probe P-Cy25S. For this analysis, boule shaped loaves (300 g) were baked, packed in polypropylene micro perforated bags and stored for 24 h at room temperature. Crust was not removed. The selected settings were as follows: test speed 1 mm/s, 30% deformation of the sample and one compression cycle. TPA [28] was carried out using Texture Analyzer TVT-XP 3.8.0.5 software (TexVol Perten Instruments, Hägersten, Sweden). Bread height, width, depth, area and specific volume were measured through the BVM-test system (TexVol Instruments). The following textural parameters were obtained: hardness (maximum peak force); fracturability (the first significant peak force during the probe compression of the bread); and resilience (ratio of the first decompression area to the first compression area). The chromaticity co-ordinates of the bread crust (Minolta CR-10 camera) were also reported in the form of a colour difference, dE *_ab_, as follows: (1)$$\mathrm{dE}\;{*}_{\mathrm{ab}}=\sqrt{{\left(dL\right)}^{2}+{\left(da\right)}^{2}+{\left(db\right)}^{2}},$$where *dL*, *da*, and *db* are the differences for *L*, *a*, and *b* values between sample and reference (a white ceramic plate having *L* = 93.4, *a* = −0.39, and *b* = 3.99). The bread crumb features were assessed after 24 h of storage using the image analysis technology with the UTHSCSA ImageTool, as previously described by Rizzello et al. [28].

The In Vitro Protein Digestibility (IVPD) of flours, sourdoughs and breads was determined by the method proposed by Akeson and Stahmann [29], with some modifications [18]. Samples were subjected to a sequential enzyme treatment mimicking the in vivo digestion in the gastro intestinal tract and IVPD was expressed as the percentage of the total protein which was solubilized after enzyme hydrolysis. The supernatant, which contained the digested proteins, was freeze-dried and used for further analyses. The modified method AOAC 982.30a was used to determine the total amino acid profile [30]. The digested protein fraction, which derived from 1 g of sample, was added of 5.7 M HCl (1 mL/10 mg of proteins), under nitrogen stream, and incubated at 110°C for 24 h. Hydrolysis was carried out under anaerobic conditions to prevent the oxidative degradation of amino acids. After freeze-drying, the hydrolysate was re-suspended (20 mg/mL) in sodium citrate buffer, pH 2.2, and filtered through a Millex-HA 0.22 µm pore size filter (Millipore Co.). Amino acids were analysed by a Biochrom 30 series Amino Acid Analyzer as described above. Because this procedure of hydrolysis does not allow the determination of tryptophan, this amino acid was estimated by the method of Pintér-Szakács and Molnán-Perl [31]. One gram of sample was suspended into 10 mL of 75 mM NaOH and shaken for 30 min at room temperature. After centrifugation (10,000 rpm for 10 min), 0.5 mL of the supernatant were mixed with 5 mL of ninhydrin reagent (1 g of ninhydrin in 100 mL of a solution HCl 37% and formic acid 96%, ratio 2:3) and incubated for 2 h at 37 °C. The reaction mixture was cooled at room temperature and made up to 10 mL with the addition of diethyl ether. The absorbance at 380 nm was measured. A standard tryptophan curve was prepared using a tryptophan (Sigma Chemicals Co.) solution in the range 0–100 μg/mL. Chemical Score (CS) estimates the amount of protein required to provide the minimal EAA pattern for adults, which was recently re-defined by FAO (Food and Agriculture Organization) in 2007 [32]. CS was calculated using the equation of Block and Mitchel [33], which compares the ratio of the individual essential amino acid (EAA) in the bread protein to that of the corresponding amino acid in the reference. The sequence of limiting essential amino acids corresponds to the list of EAA, having the lowest chemical score. The protein score indicates the chemical score of the most limiting EAA that is present in the test protein. Essential Amino Acids Index (EAAI) estimates the quality of the test protein, using its EAA content as the criterion. EAAI was calculated according to the procedure of Oser [34]. It considers the ratio between EAA of the test protein and EAA of the reference protein, according to the following equation:(2)$$EAAI=\sqrt[{n}]{\frac{\left(EAA_{1}\times 100\right)\left(EAA_{2}\times 100\right)\left(\ldots \right)\left(EAA_{n}\times 100\right)\left(sample\right)}{\left(EAA_{1}\times 100\right)\left(EAA_{2}\times 100\right)\left(\ldots \right)\left(EAA_{n}\times 100\right)\left(reference\right)}}.$$

The biological value (BV) indicates the utilizable fraction of the test protein. BV was calculated using the equation of Oser [34]: BV = ((1.09 × EAAI) − 11.70). The Protein Efficiency Ratio (PER) estimates the protein nutritional quality based on the amino acid profile after hydrolysis. PER was determined using the model developed by Ihekoronye [35]: PER = −0.468 + (0.454 × (Leucine)) − (0.105 × (Tyrosine)). The NI normalizes the qualitative and quantitative variations of the test protein compared to its nutritional status. NI was calculated using the equation of Crisan and Sands [36], which considers all the factors with an equal importance: NI = (EAAI × Protein (%)/100).

### 2.6. Volunteers Enrolment and Test Meal Administration

The sample size was calculated assuming a 10% difference in response quantitative perception and motility. We estimated that 30 patients would be required for the study to have 90% power and an α error of 5% (Sigmaplot v. 14.0, Systat Software, San Jose, CA, USA). Thus, 36 healthy subjects (20–31 years old; sex 18 male : 18 female; Body Mass Index, e.g., <25 kg/m^2^) were recruited at a tertiary referral center (Clinica Medica “A. Murri”, Department of Biomedical Science and Human Oncology, University of Bari). All subjects gave their informed consent and, at entry, underwent a full clinical anamnesis to exclude clinically evident diseases. Exclusion criteria were diagnosis of organic diseases, and therapies potentially influencing sensory perception or gastrointestinal motility. Volunteers did not show gastrointestinal symptoms in the three months before the trial and none had a previous history of gastrointestinal disease or surgery. The study was no-profit and approved by the Ethics Review Board of the University Hospital Policlinico in Bari (code: natural bread 2017; *n*. 122/18). All experiments were performed according to European Community and local Ethics Review Board guidelines. All experiments started at 8.00 a.m. after an overnight fast of at least 12 h. Antibiotics, probiotics or other drugs known to affect gastrointestinal motility or intestinal microbiota were prohibited from 10 days before the trial. To avoid prolonged intestinal H_2_-production due to the secondary presence of non-absorbable or slowly fermentable food, a diet, including meat, fish, eggs and olive oil, and water as drink [37] was prescribed starting from lunchtime, the day before the test. Any other fermentable food or sweet beverage was prohibited before and during the test. The test meal comprised one slice of bread (80 g) (equivalent to 220 kcal, with 8% protein, 0.5% fat, 63.5% carbohydrates, of which 56% starch), which was administered together with a glass of 160 mL of water plus 10 g of lactulose powder (Duphalac Dry^®^, Solvay Pharma, Turin, Italy). The lactulose is a non-absorbable substrate but fermentable to hydrogen (H_2_) by the resident oro-cecal microbiota. This fermentation is a crude estimate of the Oro-Cecal Transit Time (OCTT) [37]. The solution, at room temperature, was administered over 5 min in the presence of the examiner. For each bread, the total volume of the test meal, as determined by mimicking chewing through homogenization of the bread portion and 160 mL of water, was 215 mL (isovolumetric). On a different day, a standard liquid test meal (Nutridrink^®^; Nutricia, Milano, Italy) was used as a reference to analyse gastrointestinal symptoms and motility. It consisted of 200 mL liquid suspension containing 12 g (20%) protein, 11.6 g (19%) fat, and 36.8 g (61%) carbohydrates for a total of 300 kcal, 1260 kJ and 455 mOsm/L. Results obtained on Nutridrink in previous trials [38] also provide an internal validation tool and reference kinetic for the in vivo postprandial markers of gastrointestinal function. Lactulose (10 g = 15 mL Lattulac^®^, SOFAR, Trezzano Rosa, Milan, Italy) was added to the reference meal [25]. The final volume of the meal was 215 mL. 

A randomized scheme (double blinded for breads) was used to administer the three types of bread and the test meal to each subject. The meal administration to each volunteer was at three weeks-intervals.

### 2.7. Sensory Analysis, Perception of Appetite and Satiety, and Gastrointestinal Symptoms 

A semi-quantitative scale (score 0–3) was used to record specific perceptions: appearance (crust colour, attractiveness, porosity and elasticity); aroma (wheat, stale, mouldy, roasted and cereals); taste (sweet, salty, bitter, sour, cereals and aftertaste); and texture (hardness, crustiness, chewiness, adhesiveness and greasiness) [39], aiming at defining the peculiar organoleptic profiles of the breads. Before each assay, the mouth was washed with plain water. Food, drink, smoking and physical activity were prohibited before and during the assay. A quantitative Visual Analogue Scale (VAS 0–100 mm on a horizontal line) was used to mark the degree of pleasantness of overall appearance, aroma, taste and texture.

Detailed information was recorded from all volunteers regarding essential symptoms, which eventually occurred the prior three months (e.g., feeling of abdominal fullness, epigastric pain, nausea and/or vomiting, heartburn and additional gastrointestinal symptoms). Each symptom was scored semi-quantitatively on the 0–3-point scale for severity, frequency and duration (Table 1). The VAS monitored symptoms, together with the perception of appetite and satiety, at 30 min-intervals during 120 min following the ingestion of the test meal. VAS time-related curves for perception of appetite and satiety and gastrointestinal symptoms, according to previous studies from our group [38,40]. The corresponding Area Under Curve (AUC) were calculated through the software NCSS10 (NCSS LLC, Kaysville, UT, USA). 

### 2.8. Gallbladder and Gastric Motility

Gallbladder and gastric motility were determined simultaneously after meal ingestion [25,26,38,41,42,43,44,45,46]. Time-dependent changes of fasting and postprandial gallbladder volumes (mL) and antral areas (cm^2^) were calculated from frozen sonograms using a Noblus ultrasound (Hitachi Medical, Tokyo, Japan) equipped with the 3.5 MHz convex transducer. Gallbladder volume and antral area were measured before (−5 and 0 min) and after meal ingestion, every 15–30 min up to 120 min. Indices of gallbladder kinetics were fasting volume (mL) and residual volume (minimum volume postprandial measured in mL and percent of fasting volume). Indices of gastric emptying were antral (basal) area (cm^2^), maximal postprandial antral area (recorded after 5 min from meal ingestion), and postprandial and minimal postprandial antral areas during the whole emptying curve. Postprandial areas were also normalized to maximal areas after subtraction of basal areas
Normalized Postprandial Area = 100 × (A_t_ − A_bas_)/(A_max_ − A_bas_),(3)where A_t_ = postprandial area at any given time; A_bas_ = basal area; and A_max_ = maximal antral area. For both gallbladder and stomach, further indices included the Area Under the emptying Curve (AUC expressed as mL × 120 min) and the half-emptying time (T_1/2_, min). In particular, T_1/2_ was calculated by linear regression analysis from the linear part of the emptying curves, thus corresponding to the time in which 50% decrease of gallbladder volume and antral area were observed.

### 2.9. Oro-Cecal Transit Time

The OCTT was determined at the same time of gallbladder and gastric motility, using the H_2_ breath technique, according to standard guidelines [37,38,42,47,48,49,50]. Samples of expired air were taken before meal and subsequently, every 10 min during 180 min after meal ingestion. A pre-calibrated, portable hydrogen sensitive electrochemical device (EC60-Gastrolyzer; Bedfont Scientific, Medford, NJ, USA) was used to measure the time-dependent changes of H_2_ in breath, as a marker of cecal fermentation of the unabsorbed lactulose. Results were expressed as H_2_-excretion in parts per million (ppm). The accuracy of the detector was ± 2 ppm. An increase of 10 ppm above the baseline for two consecutive measurements was the OCTT, and calculated in min. 

### 2.10. Blood Analyses

To assess the impact of the test meal on glucose and amino acid levels, plasma samples (2 mL) were taken from venous blood of volunteers at time 0 (baseline in the fasting state) and at 30 min-intervals during 120 min after meal ingestion. For serum glucose analysis, 1 drop of blood (ca. 1 µL) was assayed using the Accu-Chek Active Meter (Roche Diagnostics, Indianapolis, Indiana, USA), and a reflectometric glucose meter with plasma-corrected results. For amino acid analysis, venous blood was collected in EDTA monovettes, and plasma was immediately separated by centrifugation (4000 rpm for 8 min at 4 °C) and stored at −20 °C. Total free amino acids were determined through the Biochrom 30 series Amino Acid Analyzer [19,51].

### 2.11. Statistical Analysis

Data from biochemical analyses were subject to one-way Analysis of Variance (ANOVA), using the IBM SPSS Statistics 26 (IBM Corporation, New York City, NY, USA) software. Data from the in vivo challenge were subjected to the ANOVA followed by the post-hoc comparison test., through the software NCSS10 (NCSS LLC, Kaysville, UT, USA). Motility parameters data were subjected to ANOVA, followed by the Tukey–Kramer Multiple-Comparison Test. A two-sided probability (*p* < 0.05) was considered statistically significant. 

## 3. Results

### 3.1. Semi-Industrial Manufacture of Sourdough and Baker’s Yeast Breads 

The traditional and most diffused protocol for making type-I sourdough (S) was used [20]. The selection of well-known and well-characterized species of sourdough lactic acid bacteria and yeasts ensured the reproducibility of the performance for bread making. Following daily refreshments, S became mature. After the last fermentation (8 h), it reached the value of pH of 3.89 ± 0.02 (Table 2). S was used to prepare two different sourdoughs. The values of pH of these sourdoughs agreed with the duration of their time of incubation. S_24_ was more acidic than S_4_. Values of TTA had an opposite trend. The cell density of lactic acid bacteria of both the sourdoughs exceeded the level of 9.0 log cfu/g. The cell numbers of S_24_ were significantly (*p* < 0.05) higher than that of S_4_. The cell density of yeasts was also the highest in S_24_. S had a concentration of lactic acid of 43.3 ± 0.3 mmol/kg. In comparison, the content of lactic acid in S_4_ was ca. 35% lower and that of S_24_ was more than twice. Almost the same differences were found for the concentration of acetic acid. The FQ ranged from 4.3 to 4.6, without significant (*p* > 0.05) differences among the sourdoughs. All the biochemical and microbiological data [20,21] proved that we produced two traditional sourdoughs (S_4_ and S_24_), having almost the same microbial load but different acidifying capabilities.

Two sourdough breads (sourdough bread, SB; and traditional sourdough bread, t-SB) and a baker’s yeast bread (BYB) were manufactured at a semi-industrial scale. BYB had the highest value of pH (5.6 ± 0.1) (Table 3). Proximal composition of the breads (Appendix A) did not show (*p* > 0.05) significant differences among macronutrients. Gluten in BYB was 8.9 ± 0.2%, and not significant (*p* > 0.05) differences were found for SB (8.7 ± 0.3%). A significant (*p* < 0.05) lower gluten amount was found in t-SB (7.6 ± 0.2%). It was previously observed that proteolysis by sourdough lactic acid bacteria on gluten proteins causes the release of soluble degradation products [18].

As expected, the acidifying activity of sourdough lactic acid bacteria during dough fermentation caused a marked decrease of the pH. t-SB had pH of 4.4 ± 0.1, while SB, which was fermented with S_4_ for a shorter time (1.5 vs. 4 h), had a significantly (*p* < 0.05) higher value of pH (4.9 ± 0.1). In agreement, the concentration of lactic acid of BYB was very low (1.7 ± 0.1 mmol/kg). The content of lactic acid of t-SB was ca. four-times higher than that found in SB. Acetic acid was found in both sourdough breads, being significantly (*p* < 0.05) higher in t-SB. The values of FQ were 4.2 ± 0.1 and 4.5 ± 0.2 for SB and t-SB, respectively. Total FAA were found at the lowest level in BYB. The concentrations of FAA of SB and t-SB were, respectively, ca. two- and two and half-times higher. 

The different total FAA concentration can be considered as an index of the degree of proteolysis of the breads, while the calculation of the protein concentration based on the organic nitrogen (Kjeldahl method) did not provide information about the degradation status of the proteins, therefore resulting similar for the three samples (Appendix A). 

During fermentation, the volume of all breads increased (ca. three-times) (Table 2). Compared to BYB, the increase for SB was slightly but significantly (*p* < 0.05) higher. The lowest volume increase was that of t-SB. In agreement, the specific volume and the gas cells area were the highest and the lowest for SB and t-SB, respectively. Intermediate were the values of BYB. SB also showed the lowest value of hardness. t-SB had the highest hardness and the lowest fracturability, which corresponded to a remarkable friability of the crumb. The instrumental colour analysis indicated small differences for lightness (L) and colorimetric coordinates between BYB and SB. t-SB had the lowest and highest values of L and ΔE, respectively. All biochemical and textural data proved that we manufactured representative sourdough and baker‘s yeast breads, with differences between sourdough breads (SB and t-SB) that mimicked regional traditions [20,21]. 

### 3.2. In Vitro Protein Digestibility 

The value of IVPD for BYB was ca. 63.7% (Table 3). Overall, the sourdough fermentation led to an increase of the in vitro protein digestibility: ca. 8 and 16% for SB and t-SB, respectively. Lys, Met and Trp were the limiting amino acids of the three types of bread. Nevertheless, the protein score was the highest for SB and t-SB. Similarly, the values of EAAI, BV and PER, which are used to estimate the quality of food proteins, were significantly (*p* < 0.05) higher in SB with respect to BYB. t-SB had the highest values of these indices. The Nutritional Index (NI) calculation referred to both the amount of digestible protein fraction and the ratio of essential amino acids. SB and, especially, t-SB had values of this index ca. 21 and 62% higher than that determined for BYB. As documented in the literature [18,52], we confirmed that the in vitro protein digestibility improves with sourdough fermentation.

### 3.3. Breads Sensory Profile and Perception of Appetite, Satiety, and Gastrointestinal Symptoms

Volunteers used questionnaires to describe the sensory properties of the three types of bread. They explicited attributes for appearance, aroma, taste and texture in a semi-quantitative scale. No significant (*p* > 0.05) differences were found for attributes of appearance, such as attractiveness and elasticity (Figure 1A). Only the crust colour and porosity of SB were scored as significantly (*p* < 0.05) the highest. The scores for stale and mouldy aroma were low for all the three types of bread. Wheat aroma primarily connoted BYB. The aroma of SB was described as roasted and cereals. Bitter and sour taste mainly distinguished t-SB, with the same attributes that received significantly (*p* < 0.05) lower scores for SB. Overall, the aftertaste of both the sourdough breads was the most intense. Volunteers did not perceive significant (*p* > 0.05) differences for hardness, crustiness and greasiness among the three types of bread. Chewiness and adhesiveness scores were the highest in BYB. Volunteers assessed the bread pleasantness according to the Visual Analogic Scale (VAS) approach. The overall taste and texture did not significant (*p* > 0.05) differ (Figure 1B).

Compared to the other two types of bread, the overall appearance and aroma were significantly (*p* < 0.05) higher for SB. Using VAS approach, appetite and satiety, and the gastrointestinal symptoms were monitored during 120 min following ingestion. After 30 min, SB and t-SB stimulated more appetite than BYB (Figure 2A).

After 60 min, no significant (*p* > 0.05) differences were found among breads. Sixty minutes after ingesting Nutridrink (NU), the significantly (*p* < 0.05) lowest Area Under Curve (AUC) indicated the lowest perception of appetite. After 120 min from ingestion, the highest satiety perception was associated with NU consumption. Comparing breads, the ingestion of t-SB was associated with the curve having the significantly (*p* < 0.05) highest AUC. No symptoms of nausea were perceived (scores always lower than 10 mm) after ingesting breads (Figure 2B). Significantly (*p* < 0.05) higher scores were reported after NU ingestion. The consumption of NU also associated to the lowest fullness perception. AUC for fullness were similar among breads. Nevertheless, t-SB showed the highest score at 30 min. Volunteers did not report epigastric pain after ingesting breads or NU. In particular, VAS scores were lower than 15 mm during the 120 min following the samples ingestion. AUC was similar (*p* > 0.05) and lower than 1000 for all the breads and NU.

### 3.4. Gallbladder and Gastric Motility and Determination of the OCTT

Volunteers showed similar fasting gallbladder volumes across the four days of challenging (17.6 ± 1.2 − 18.9 ± 1.3 mL) (Figure 3A and Appendix A). 

The ingestion of the three types of bread induced similar responses. In detail, the mean gallbladder ejection volume ranged from 35.5 to 43.5% (mean residual volume of ca. 11 mL), the half-emptying time varied from 26.7 ± 2.3 to 33.6 ± 2.8 min, and the half-refilling rate was 0.1 ± 0.0 mL/min. Compared to breads, NU induced a faster gallbladder emptying and refilling. This was confirmed by significantly (*p* < 0.05) different kinetic parameters (ejection volume of 68.5 ± 3.1%; residual volume of 5.9%; half-emptying time of 21.9 ± 1.4 min; and half-refilling rate of 2.4 ± 0.4 mL/min). At baseline, gastric basal antral area ranged from 4.0 ± 0.2 to 4.3 ± 0.3 cm^2^ (Figure 3B). Within five minutes after ingestion, the antral area reached the maximum postprandial area (10.7 ± 0.4 − 11.7 ± 0.5 cm^2^), without any significant (*p* > 0.05) difference among breads. The ingestion of different meals corresponded to different time-related decreases of the antral area. In particular, the emptying rate corresponding to SB and t-SB ingestion was significantly (*p* < 0.05) lower than that of BYB (−0.24 ± 01 and −0.28 ± 0.1 vs. −1.20 ± 0.1 cm^2^/min, Appendix A). Accordingly, the half-emptying time of BYB (43.7 ± 4.4 min) was significantly (*p* < 0.05) longer than that found for SB and t-SB (34.8 ± 2.4 and 30.8 ± 2.3 min, respectively). The AUC amplitude for BYB was the highest (854 ± 119 vs. 774 ± 105 and 758 ± 105 cm^2^ × 120 min for SB and t-SB, respectively) (Appendix A). Response to NU ingestion showed that the gastric half-emptying time was markedly and significantly (*p* < 0.05) lower than that of BYB. This probably because of the different viscosity after chewing and ingestion. Nevertheless, it was comparable to both sourdough breads, having crumbly texture and lower fracturability than BYB. In detail, the NU ingestion was characterized by emptying rate, half-emptying time and AUC of −0.66 ± 0.0, 29.0 ± 1.9 min and 765 ± 19 cm^2^ × 120 min, respectively. The OCTT was assessed through the H_2_ breath technique. Volunteers consumed lactulose together with breads and the fermentation of this indigestible sugar by colon microbes produced H_2_. The concentration of H_2_ in breath was measured every 10 min after meal ingestion. The time needed to exceed the baseline value (10 ppm) was determined and considered as the entrance point of the food bolus into the large intestine, which corresponds to OCTT. The OCTT median value for BYB was 89.5 min, which was not significantly (*p* > 0.05) different than that found for SB (80.5 min). On the contrary, this value was markedly and significantly (*p* < 0.05) longer than that of t-SB (69.5 min) (Figure 4A,B). 

The OCTT (median value 103.2 ± 3.4 min) of NU was significantly (*p* < 0.05) the lowest. The H_2_ peak for BYB was up to 30% higher (*p* < 0.05) compared to both sourdough breads and NU (Figure 4C).

### 3.5. Postprandial Glycaemia and Free Amino Acids Absorption 

Glycaemia was assessed by analysing the concentration of glucose from blood samples that were collected after meal ingestion. All breads showed peaks of serum glucose concentration from 40 to 60 min after ingestion. The peak following BYB ingestion corresponded to 14.3 ± 0.39 mg/mL (Figure 5A), which was ca. 5% lower (*p* < 0.05) than those reported for SB and t-SB. Compared to BYB, SB and t-SB generated a lower glycaemic curve (Figure 5B). 

The AUC of BYB was 14947 ± 416 mg × 120 min, while the areas for SB and t-SB were 11 and 25% lower (*p* < 0.05), respectively. Before meal ingestion, the concentration of total FAA in the volunteer blood plasmas ranged from 45 ± 5 to 60 ± 3 mg/L. This concentration increased after ingesting all breads, and reached the maximum peak at 60 min. At this time, the concentrations (median values) were 145 ± 6, 152 ± 5 and 190 ± 5 mg/L for blood plasmas collected after ingesting BYB, SB and t-SB, respectively (Figure 6A). 

Although the concentrations of total FAA of both sourdough breads were higher, the differences among breads were not significant (*p* > 0.05). After 60 min, the concentrations of total FAA decreased according to kinetics that varied depending on the type of bread (Figure 6B). At 120 min, the concentration of total FAA of BYB was similar (*p* > 0.05) to the baseline value (decrease by 67%). On the contrary, the decreases observed in the blood plasmas of volunteers who ingested SB and t-SB were very low (17 and 10%, respectively). 

## 4. Discussion

Scientific evidence has been gathered to prove the nutritional effectiveness of the sourdough fermentation [8,53] but investigations on bread digestibility are still too limited and partial. The tradition and popular opinion converge on the alleged greater digestibility and lightness of long-time fermented sourdough breads with respect to fast processing by chemicals or baker’s yeast. First, we aimed at deepening the knowledge in this regard, combining wide-spectrum characterization of breads, in vitro nutritional indices, and, primarily, in vivo postprandial markers from healthy volunteers. 

All microbiological (e.g., cell densities and ratio between lactic acid bacteria and yeasts of 100:1) and biochemical (e.g., pH, content of organic acids, FQ and concentration of total FAA) data supported the peculiar features of baker´s yeast bread and two different types of sourdough breads [20]. 

A number of hedonistic attributes (e.g., palatability, aroma, texture, appearance and overall pleasantness) markedly influence the consumer approach to a meal. These attributes relate to food chemical and physical features, which have shaped during processing [38]. Sourdough fermentation causes the appearance of distinctive sensory attributes in breads [7]. Salty, bitter and, especially, sour tastes distinguished sourdough breads, with very low scores for chewiness and adhesiveness. As usual for dietary studies, the assessment of pleasantness was through the VAS [13,54]. Reflecting the current market trend and considering the relatively young age (20–31 years old) of volunteers, SB, manufactured with a mild acidification, was preferred for aroma and appearance (Figure 1). The most pronounced sour taste primarily attained to the old tradition. Together with the sensory attributes, the perception of appetite, satiety and gastrointestinal symptoms is essential to assess food digestibility because tightly correlated with gastric emptying and intestinal fermentation [13,14]. It must be highlighted that the present study aimed at investigating bread digestibility in healthy subjects, acting as reference group for further studies on subjects with upper GI symptoms such as bloating, reflux, epigastric pain or irritable bowel syndrome. Absent or low intensity gastrointestinal symptoms, including fullness, are associated to post-prandial well-being [13]. As expected, the ingestion of all the three types of bread did not cause higher nausea and epigastric pain. SB, made with moderate sourdough acidification, stimulated more the appetite. Within 60 min after ingestion, SB also induced lower satiety and fullness perceptions (Figure 2). Other authors [13,55] described an inverse correlation between appetite and satiety perceptions. t-SB, having the most intense acidic and salty taste, and the most compact structure, induced the highest fullness perception in the shortest time (30 min). A previous report [13] on croissants made with sourdough showed almost similar results, combining lower appetite and higher satiety with respect to the consumption of croissants made with baker’s yeast.

NU is a commercial preparation with a defined nutrient composition and gastric and small intestinal motility [41,42]. The gallbladder response did not differ among breads (Figure 3B), nevertheless, the response to the isovolumetric NU was different. As expected, compared to breads, the ingestion of NU caused a markedly higher ejection volume and refilling time, with a kinetic curve of the gallbladder volume that determined an AUC ca. two-times higher. The higher content of lipids of NU might had been responsible for these differences.

Overall, the nutrient composition and quantity of the ingested food influence its stomach transit. The alteration of the transit could induce fullness and nausea [13]. As estimated using paracetamol adsorption or the ^13^C-octanoic acid breath test, previous in vivo challenges [56,57] failed to show differences in gastric emptying after ingesting breads made with different cereal flours and leavening agents. Through an ultrasonographic analysis at gastric level, we were able to show a faster emptying time for sourdough breads compared to BYB (Figure 3A). Sourdough breads had lower values of pH and higher TTA than baker’s yeast bread, which led to the hypothesis of a quicker gastric acidification of the bolus. The different status of the organic nitrogen fraction among breads might had played also a relevant function. Contrary to yeasts, sourdough lactic acid bacteria have a more intense proteolytic activity because of an efficient proteolytic system, comprising cell-wall associated proteinases, a wide portfolio of intracellular peptidases and specific membrane transporters [8]. The progressive hydrolysis of wheat naive proteins, including gluten, led to a consistent increase of peptides and free amino acids during sourdough fermentation. Like in a process of pre-digestion, the occurring proteolysis increased the amount of the in vitro digestible protein fraction (up to ca. 80%) as shown by several nutritional indices (IVPD, EAAI, BV, PER and NI). The degree of proteolysis was proportional to the duration of the fermentation. As analysed by magnetic resonance imaging, faster gastric emptying also followed the ingestion of sourdough croissants with respect to those made with baker’s yeast [13]. All these data contrasted previous findings of Liljeberg and Bjorck [58] that suggested a decrease of the gastric emptying in presence of organic acids. The rapidity of the gastric emptying correlates with a reduced perception of nausea, abdominal discomfort, fullness and satiety [59]. This correlation perfectly fits with what observed for SB but disagreed with respect to the behaviour of t-SB, which had comparable gastric emptying rate but caused a more persistent fullness and satiety perception. The hypothesis is that sourdough baked goods did not induce a mechanical satiety but modulated the feelings of hunger and satiety by stimulating the hormonal response through metabolically active compounds [13]. The type and amount of proteins influence the gastric emptying time, and the satiety and appetite perceptions through the modulated release of hormones, such as glucagon-like peptide (GLP)-1, oxyntomodulin, pancreatic polypeptides and glucose-dependent insulin-trophic polypeptide (GIP) [60,61]. Although it was demonstrated that an increased protein content delayed gastric emptying [60], no information on the effect of hydrolysed proteins is currently available.

The administration of lactulose together with the test meal is one of the most efficient tools to monitor the oro-cecal transit time through the estimation of the H_2_ level in breath [35]. The dose of lactulose administered to volunteers is markedly lower than the amount previously associated to the gastrointestinal motility increase and osmotic diarrhoea [62]. First, we showed a prolonged transit time for the baker’s yeast bread and a faster passage of sourdough breads, especially when made with traditional and long-time fermentation, whose transit lasted ca. 20 min less than BYB (Figure 4). This finding was consistent with the results from gastric emptying. It was hypothesized a different carbohydrate profile between baked goods made with baker’s yeast and those fermented with sourdough. The diverse and complementary metabolic activities of the sourdough microbiota, together with flour endogenous enzymes (e.g., α- and β-amylases, gluco-amylases), led to a substrate composition that uniquely influenced the gas production by intestinal microbes [13]. An in vitro study [63] demonstrated that the pattern of carbohydrates derived from sourdough fermentation caused a low cumulative gas emission after 15 h of fermentation by the intestinal microbes. 

Differences in carbohydrate digestibility and absorption among breads determined different post-prandial glycaemia responses (Figure 5). Sourdough breads had the lowest values. The glycaemic index (GI) (ratio of the areas under the glycaemic curves between tested bread and reference white bread, in this case BYB) was ca. 89 and 75 for SB and t-SB, respectively. These findings confirmed other in vivo determinations [16,17,64]. Primarily, the decreased GI of sourdough breads correlated to biological acidification [65] in presence of low values of pH (3.5–4.0), which promote the formation of resistant starch [66]. Fibbers [16] and phenols [67] may have an effect. During sourdough fermentation, lactic acid bacteria have the capability to increase the fiber soluble fraction and the concentration of free phenolic compounds [8,52]. Nevertheless, because of the low concentration of fibres and phenols in the white wheat flour used for bread making, a weak contribution is conceivable. Because the relevant difference for proteolysis between baker’s yeast and sourdough breads, we innovatively estimated the total FAA absorption as a further measurement of the bread digestibility. The total concentration of FAA in blood plasmas increased following the ingestion of all breads. After ingestion of sourdough breads, having a concentration of total FAA markedly higher than that of BYB, the levels of these nitrogen derivatives in blood plasmas maintained high and almost constant for extended time (Figure 6). 

## 5. Conclusions

As a staple food, information on bread digestibility deserve a marked interest from a nutritional perspective. Bread is the generic name for a group of leavened food products, manufactured with a wide range of flours and using different leavening processes. The traditional and popular opinion concerns a high digestibility of long-time fermented breads although confirmatory scientific data are lacking. The present study presents some limitations: (i) only healthy normal subjects (without gastrointestinal symptoms) were enrolled; (ii) questionnaires for gastrointestinal symptoms investigation are not largely validated (iii) H_2_-breath test (with lactulose) is not the gold-standard method to asses gastrointestinal transit, however it is non-invasive, well accepted by subjects (radiations or intubations are not required), and largely used to assess the OCTT. Nevertheless, we combined wide-spectrum characterization of breads, in vitro nutritional indices, and in vivo postprandial markers, which all agreed and somewhat explained the presumed better digestibility of sourdough compared to baker’s yeast breads. Moreover, a different clinical response to sourdough breads, representative of the most common production protocols and characterized by different levels of acidification and proteolysis, was highlighted.

## Figures and Tables

**Figure 1 nutrients-11-02954-f001:**
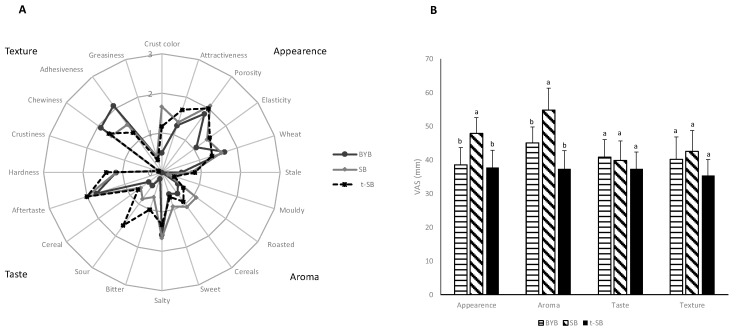
Sensory analysis of the three types of bread. BYB, baker´s yeast bread leavened with baker’s yeast (1.5%, *w*/*w*) for 90 min at 30 °C; SB, sourdough bread leavened with S_4_ (20%, *w*/*w*) and baker’s yeast (1.5%, *w*/*w*) for 90 min at 30 °C; t-SB, sourdough bread leavened with S_24_ (20%, *w*/*w*) for 4 h at 30°C. (**A**) spider web chart of the perception scores (semi-quantitative scale 0–3); (**B**) degree of pleasantness (visual analogue scale, VAS, 0–100 mm). ^a–c^ Within the same parameter, values with different superscript letters differ significantly (*p* < 0.05).

**Figure 2 nutrients-11-02954-f002:**
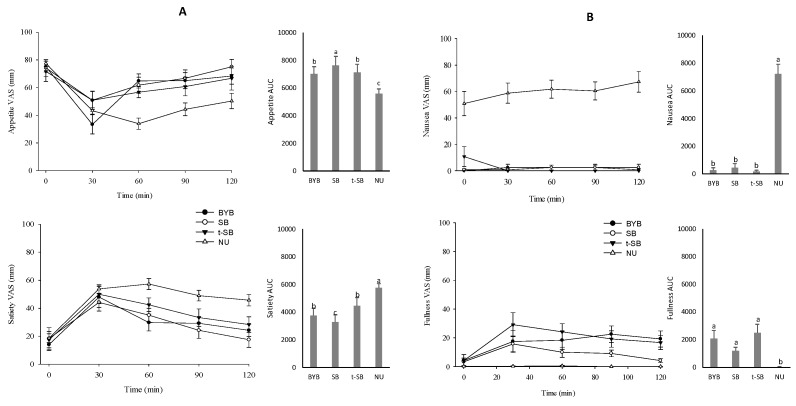
Perception of appetite and satiety (**A**), and gastrointestinal symptoms (nausea and fullness) (**B**) in response to ingestion of the test meals. Time-dependent changes were scored with Visual Analogic Scale (VAS, 0–100 mm), and represented as mean ± SEM (Standard Error of the Mean) and area under curve (AUC). BYB, baker´s yeast bread leavened with baker’s yeast (1.5%, *w*/*w*) for 90 min at 30 °C; SB, sourdough bread leavened with S_4_ (20%, *w*/*w*) and baker’s yeast (1.5%, *w*/*w*) for 90 min at 30 °C; t-SB, sourdough bread leavened with S_24_ (20%, *w*/*w*) for 4 h at 30 °C. NU, Nutridrink used as the reference. ^a–c^ Within the same parameter, values with different superscript letters differ significantly (*p* < 0.05).

**Figure 3 nutrients-11-02954-f003:**
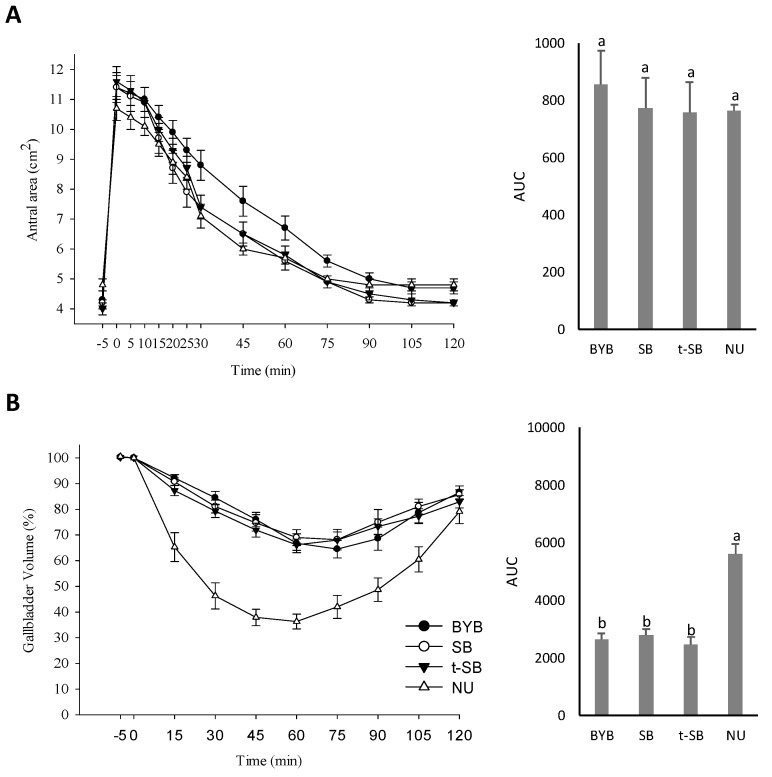
Gastric (**A**) and gallbladder (**B**) emptying curves in response to the ingestion of the test meals (isovolumetric). Time-dependent changes of antral area (cm^2^) are represented as mean ± SEM (Standard Error of the Mean) and area under curve (AUC). Time-dependent changes of gallbladder volume (mL) are represented as mean ± SEM and area under curve (AUC). BYB, baker´s yeast bread leavened with baker’s yeast (1.5%, *w*/*w*) for 90 min at 30 °C; SB, sourdough bread leavened with S_4_ (20%, *w*/*w*) and baker’s yeast (1.5%, *w*/*w*) for 90 min at 30 °C; t-SB, sourdough bread leavened with S_24_ (20%, *w*/*w*) for 4 h at 30 °C. NU, Nutridrink used as the reference. ^a–b^ Within the same parameter, values with different superscript letters differ significantly (*p* < 0.05).

**Figure 4 nutrients-11-02954-f004:**
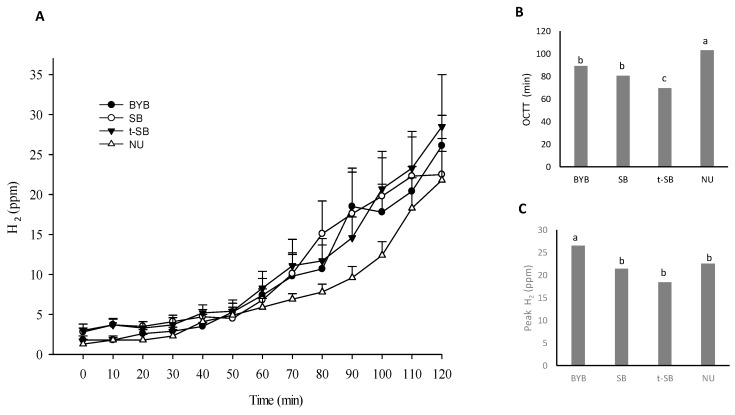
Oro-cecal transit time (OCTT) in response to the ingestion of test meals. (**A**) time-dependent curves of H2 concentration (ppm) in exhaled air, expressed as mean ± SEM (Standard Error of the Mean); (**B**) OCTT (min); (**C**) H2 peak. BYB, baker´s yeast bread leavened with baker’s yeast (1.5%, *w*/*w*) for 90 min at 30 °C; SB, sourdough bread leavened with S_4_ (20%, *w*/*w*) and baker’s yeast (1.5%, *w*/*w*) for 90 min at 30 °C; t-SB, sourdough bread leavened with S_24_ (20%, *w*/*w*) for 4 h at 30 °C. NU, Nutridrink used as the reference. ^a–c^ Within the same parameter, values with different superscript letters differ significantly (*p* < 0.05).

**Figure 5 nutrients-11-02954-f005:**
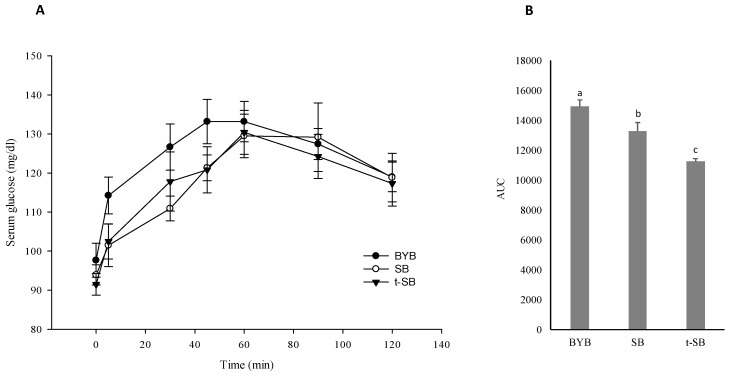
Serum glucose concentration in response to the ingestion of the test meals. (**A**) time-dependent curves of serum glucose (mg/dL) and (**B**) area under curve (AUC). Data are expressed as mean ± SEM (Standard Error of the Mean). BYB, baker´s yeast bread leavened with baker’s yeast (1.5%, *w*/*w*) for 90 min at 30 °C; SB, sourdough bread leavened with S_4_ (20%, *w*/*w*) and baker’s yeast (1.5%, *w*/*w*) for 90 min at 30 °C; t-SB, sourdough bread leavened with S_24_ (20%, *w*/*w*) for 4 h at 30 °C. ^a–c^ Values with different superscript letters differ significantly (*p* < 0.05).

**Figure 6 nutrients-11-02954-f006:**
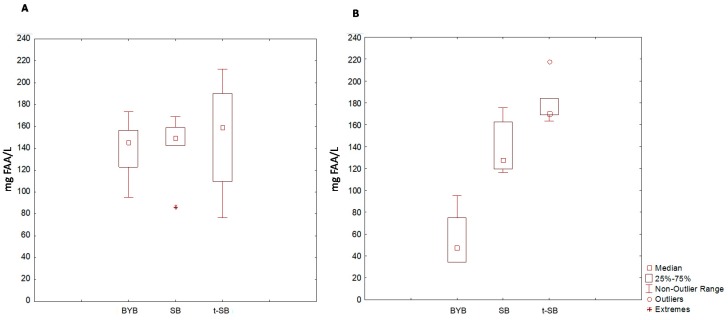
Serum total free amino acids (FAA) concentration in response to the ingestion of test meals. Aggregated data are represented in box-plots: (**A**) total FAA concentration at 60 min after ingestion; (**B**) total FAA concentration at 120 min after ingestion. BYB, baker´s yeast bread leavened with baker’s yeast (1.5%, *w*/*w*) for 90 min at 30 °C; SB, sourdough bread leavened with S_4_ (20%, *w*/*w*) and baker’s yeast (1.5%, *w*/*w*) for 90 min at 30 °C; t-SB, sourdough bread leavened with S_24_ (20%, *w*/*w*) for 4 h at 30 °C.

**Table 1 nutrients-11-02954-t001:** Clinical characteristics of enrolled volunteers.

*n*	36
Males:Females	18:18
Age years (range)	25 ± 1.1 (20–31)
BMI, kg/m^2^ (range)	22.4 ± 0.9 (18.6–29.4)
Fullness (severity) *	0.8 ± 0.2
Fullness (frequency) **	0.7 ± 0.2
Fullness (duration) ***	0.7 ± 0.2
Epigastric pain (severity)	0.9 ± 0.4
Epigastric pain (frequency)	0.6 ± 0.3
Epigastric pain (duration)	0.6 ± 0.3
Nausea/vomiting (severity)	0.2 ± 0.2
Nausea/vomiting (frequency)	0.2 ± 0.1
Nausea/vomiting (duration)	0.6 ± 0.3
Heartburn (severity)	0.5 ± 0.2
Heartburn (frequency)	0.5 ± 0.2
Heartburn (duration)	0.4 ± 0.2

BMI, body mass index; data are expressed as mean ± SEM; Severity, frequency and duration are expressed on a 0–3 semiquantitative score (* 0 = absent; 1 = mild; 2 = moderate; 3 = severe; ** 0 = none; 1 = sometime; 2 = often; 4 = interference with daily activities; *** 0 = none; 1 = less than 30 min daily; 2 = between 30 min and 2 h daily; 3 = more than 2 h/daily).

**Table 2 nutrients-11-02954-t002:** Biochemical and microbiological characteristics of sourdoughs (dough yield of 160) used for bread making.

	S	S_4_	S_24_
pH	3.89 ± 0.02 ^b^	4.26 ± 0.01 ^a^	3.55 ± 0.01 ^c^
TTA Total titratable acidity (mL NaOH 0.1M/10g)	6.60 ± 0.11 ^b^	3.20 ± 0.09 ^c^	9.00 ± 0.15 ^a^
LAB (Log cfu/g)	9.53 ± 0.13 ^b^	9.18 ± 0.10 ^b^	9.75 ± 0.08 ^a^
Yeasts (Log cfu/g)	7.33 ± 0.05 ^b^	7.07 ± 0.07 ^c^	7.50 ± 0.05 ^a^
Lactic acid (mmol/kg)	43.3 ± 0.3 ^b^	28.2 ± 0.2 ^c^	93.3 ± 0.3 ^a^
Acetic acid (mmol/kg)	10.2 ± 0.2 ^b^	6.2 ± 0.2 ^c^	20.3 ± 0.2 ^a^
Fermentation Quotient (FQ)	4.3 ± 0.2	4.5 ± 0.2	4.6 ± 0.2
Total Free Amino Acids (g/kg)	0.722 ± 0.018 ^c^	1.380 ± 0.024 ^b^	7.174 ± 0.032 ^a^

S, type-I sourdough, which was mature after four refreshments; S_4_ and S_24_, sourdoughs produced by mixing S with wheat flour and water, and fermented at 30°C for 4 and 24 h, respectively. ^a–c^ Values in the same row with different superscript letters differ significantly (*p* < 0.05).

**Table 3 nutrients-11-02954-t003:** Chemical, technological, and nutritional characteristics of the three types of bread.

	BYB	SB	t-SB
	Dough (before baking)		
pH	5.6 ± 0.1 ^a^	4.9 ± 0.1 ^b^	4.4 ± 0.1 ^c^
TTA (mL NaOH 0.1m/10g)	3.0 ± 0.1 ^c^	5.0 ± 0.2 ^b^	6.9 ± 0.2 ^a^
Lactic acid (mmol/kg)	1.7 ± 0.1 ^c^	7.2 ± 0.1 ^b^	26.4 ± 0.3 ^a^
Acetic acid (mmol/kg)	nd	1.4 ± 0.1 ^b^	5.8 ± 0.1 ^a^
FQ	nd	4.2 ± 0.1 ^b^	4.5 ± 0.2 ^a^
Total FAA (g/kg)	0.70 ± 0.02 ^c^	1.33 ± 0.02 ^b^	1.71 ± 0.02 ^a^
	Bread		
Volume increase (%)	231 ± 5 ^b^	245 ± 7 ^a^	223 ± 4 ^c^
Specific volume (cm^2^/g)	3.3 ± 0.1 ^b^	3.6 ± 0.1 ^a^	2.9 ± 0.1 ^c^
Textural parameters			
Hardness (g)	3210 ± 10 ^b^	3150 ± 21 ^c^	3472 ± 11 ^a^
Resilience	0.85 ± 0.02 ^a^	0.81 ± 0.05 ^a^	0.72 ± 0.02 ^b^
Fracturability (g)	3075 ± 5 ^a^	2956 ± 10 ^b^	2282 ± 10 ^c^
Image analysis			
Black pixel area (%)	44.0 ± 1.8 ^b^	52.6 ± 2.2 ^a^	44.4 ± 1.5 ^b^
Color analysis			
L	60.2 ± 1.6 ^a^	60.4 ± 0.7 ^a^	54.7 ± 0.8 ^b^
A	10.4 ± 0.8 ^b^	9.3 ± 0.6 ^b^	11.7 ± 0.2 ^a^
B	35.7 ± 1.0 ^a^	34.1 ± 0.2 ^a^	33.6 ± 0.5 ^b^
ΔE	48.1 ± 1.7 ^b^	46.6 ± 0.6 ^b^	51.3 ± 0.6 ^a^
Nutritional indexes			
In vitro protein digestibility (IVPD, %)	63.7 ± 1.2 ^c^	71.6 ± 1.1 ^b^	79.8 ± 1.4 ^a^
Limiting Amino Acids	Lysine Methionine Tryptophan	Lysine Methionine Tryptophan	Lysine Methionine Tryptophan
Protein score (%)	18.2 ± 0.2 ^c^	24.2 ± 0.2 ^b^	59.2 ± 0.5 ^a^
Essential amino acids Index (EAAI)	43.2 ± 0.4 ^c^	56.3 ± 0.5 ^b^	72.3 ± 0.5 ^a^
Biological value (BV)	35.4 ± 0.3 ^c^	39.7 ± 0.3 ^b^	54.1 ± 0.4 ^a^
Protein Efficiency Ratio (PER)	21.1 ± 0.2 ^c^	23.5 ± 0.2 ^b^	53.5 ± 0.3 ^a^
Nutritional Index (NI)	2.8 ± 0.1 ^c^	3.4 ± 0.1 ^b^	5.5 ± 0.1 ^a^

BYB, baker´s yeast bread leavened with baker’s yeast (1.5%, *w*/*w*) for 90 min at 30 °C; SB, sourdough bread leavened with S_4_ (20%, *w*/*w*) and baker’s yeast (1.5%, *w*/*w*) for 90 min at 30 °C; t-SB, sourdough bread leavened with S_24_ (20%, *w*/*w*) for 4 h at 30 °C. ^a–c^ Values in the same row with different superscript letters differ significantly (*p* < 0.05); nd: not detected.

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
