# Peer review of "Sourdough Fermented Breads are More Digestible than Those Started with Baker’s Yeast Alone: An In Vivo Challenge Dissecting Distinct Gastrointestinal Responses"

_nutrients, 2019, doi:10.3390/nu11122954_

Round 1

Reviewer 1 Report

This paper investigates the digestibility and tolerance if sourdough fermented breads compared to those started with baker’s yeast alone. This is innovative work that is very interesting, timely, and much needed. However, the design of the in vivo trial is not adequate to answer whether sourdough bread is better digested or tolerated due to few major limitations.

GENERAL

There are several grammatical errors throughout the manuscript – please go through it and correct them.

INTRODUCTION

Line 47: explain what leavening agents do Line 57: explain what back slopping is Line 61: provide a reference for the claim made Line 64-65: “using…processing” – explain why this is important Line 69: what does “positively influence” mean – please elaborate. Line 72: the claim “satisfy the consumer expectations” is not supported by the reference provided (9) – please provide a reference that proves this claim.

METHODS

Was the nutritional composition of the 3 breads assessed (energy, macronutrients, micronutrients)? Were gluten and oligosaccharides assessed? If so, provide the data. Line 16: based on what preliminary data was the sample size calculation performed? Provide reference. MAJOR: Participants of the study were healthy with no GI symptoms. This poses a question on whether the tolerance and digestibility of the bread can be measured in this cohort since they are not the ones who report issues with digesting bread. Would recruiting participants with GI symptoms been more suitable? MAJOR: A major limitation of the study is the provision of lactulose, a poorly absorbed sugar, along with the test bread. Lactulose is well known to impact gut function and symptoms and therefore is a major confounder for the results since it was taken with the test bread GI motility: there needs to be acknowledgment that H2 is not gold standard. Gold standard methods are for example the radio-opaque marker technique. Line 240-41: please explain when the liquid meal test was given exactly – was it immediately after the test bread consumption? Line 245: how was randomisation performed? Lines 243-62: it appears that outcomes have been measured using two separate methods: a 4 scale Likert scale, and VAS scales. Is this correct? If so, why was this performed? This increases Type I statistical error. MAJOR: The measurement technique for measuring GI symptoms is not validated. This needs to be acknowledged. Line 258: Since digestibility is a key aspect of your study, these data should be provided in the main paper, not on the Supplementary Table.

RESULTS

In general, I believe the strong conclusions made throughout the results are not representative of the methodology of this study. I suggest it is reworded to better represent the quality of the data. Some examples:

Lines 333-35: please reword the sentence as at the moment it is difficult to understand. MAJOR: Line 366: I believe this title overinflates the actual results. Please amend. Line 400: “appreciated” – this is too vague – what outcome was this based on? Line 409: again, since GI symptoms are key to this study, data should be shown. MAJOR: Line 410: Again, this title overinflated the actual results. Please amend. Importantly, the authors did not measure gastrointestinal transit! They measured oro-ceacal transit! Line 447: since this was not statistically significant, it cannot be claimed it was “lower”. Line 461-62: Again, this title overinflated the actual results. Please amend.

DISCUSSION

In general, I believe the strong conclusions made throughout the discussion are not representative of the methodology and the results of this study. I suggest it is reworded to better represent the quality of the data. Some examples:

Line 496: “functional” – do you mean mechanistic insights? Lines 503-514: this is methodology. No need to repeat in the discussion. Same with lines 538-9. Line 527: “did not cause nausea” – I believe this is an overestimation of the actual results. Causality cannot be established. Perhaps replace with “did not lead to higher nausea…” MAJOR: Lines 528-29: I do not think this study can be used to infer potential effectiveness for functional bowel disorders, since the pathophysiology of such disorders is complex and not simply attributed to diet. In addition. The gut physiology and function in people with functional bowel disorders is different to healthy people; therefore the results of the current study, which was performed in healthy people, cannot be applied to people with an altered gut physiology and function. Line 548: nausea and fullness is not only caused by an alteration in GTT.. MAJOR: There is no mention of any limitations of this study! There are major limitations that must be acknowledged and the authors should explain how these affect their results. The major limitations include, but are not limited, to using non-validated questionnaires, not using a gold-standard methodology for assessing GTT, providing lactulose with the test meal which is a considerable confounder, recruiting healthy people who have no GI symptoms.

CONCLUSION

In general, I believe the strong conclusions made throughout the discussion are not representative of the methodology and the results of this study. I suggest it is reworded to better represent the quality of the data. Some examples:

Line 610: this suggests that this study did provide “confirmatory” data, which is not the case. Line 612: “better digestibility” – I believe this is an overstatement, especially due to the lack of validated techniques to assess gut symptoms.

Reviewer 2 Report

The manuscript ”Sourdough fermented breads are more digestible than those started with baker’s yeast alone: an in vivo challenge dissecting distinct gastrointestinal responses” by Rizzello et al. describes the food technological and nutritional characteristics of different types of wheat brads, as well as their digestibility both in vitro and in vivo. The methodologies utilized are up-to-date, and the examination of different aspects affecting and indicating bread digestibility is thorough. In addition, fermented foods represent a very current field in nutrition research, and in vivo studies on especially protein digestibility and bioavailability are urgently needed.

However, I have some concerns. First, the approach to the subject is very wide, and I would even consider dividing the manuscript into two separate articles to enable the results to be viewed and discussed in more detail. Second, the authors have chosen to use Nutridrink as a control for the in vivo studies, and although this choice has been briefly justified in the methods  and discussion sections and the differences in the gallbladder responses have been pondered to be related with the higher lipid content of Nutridrink, I found it difficult to find the real connection between the reference product and the test products because of their very different nutritional values and characteristics. Third, for the in vivo studies, gallbladder emptying assay and a questionnaire on gastric symptoms have been applied, and it was a bit unclear to me why these aspects are important considering that, for example, nausea is not an expected outcome. I’m sure that, for instance, new knowledge on dietary effects of sourdough bread has been gained with gallbladder emptying trials, but these aforementioned approaches need more profound theoretical background to be presented. In addition, I was a little confused about the concept of appetite in this study; for example, I don’t find it very surprising that while the satiety decreases, the appetite increases, but in the manuscript, the induction of appetite was presented as an indicator of the appreciation towards the fermented breads, not as an indicator of reduced satiety.

Minor comments:

Abstract:

Not necessary to mention the reference product in the abstract. Overall, the abstract could be shortened.

Introduction:

Introduction is too long for a research article and needs to be condensed.

Page 2, Line 67: Scopus count is from September 2018 (October 2019). What is the number of publications now?

Page 2, Lines 78-79: What do you mean by saying: “bread digestibility relies on factors of different nature: the perception of appetite, satiety and gastrointestinal symptoms after ingestion,…”? How do these factors affect digestibility?

Page 2, Line 87-91: Here, you mention that the in vivo trials were conducted first and after that, the biochemical analyses were done. However, in materials and methods section an in results, in vivo tests are presented after other tests. Although a minor thing, I found this a bit distracting.

Materials & Methods:

Page 3, Lines 116 & 132: The meanings of abbreviations TTA and FAA are missing.

Results:

Headline 3.3. is a bit vague. Can you, please, rephrase?

Discussion:

Page 4, Line 540-541: I wouldn’t say that the references used here are similar interventions than the one presented in this manuscript. What has been used as a reference in previous bread studies? Is it necessary to present the Nutridrink results in this article?

Conclusions:

This section mainly repeats the general things mentioned in the beginning of the manuscript. More concrete conclusions related with this current study would be more effective.
